# Antifreeze Proteins and Their Practical Utilization in Industry, Medicine, and Agriculture

**DOI:** 10.3390/biom10121649

**Published:** 2020-12-09

**Authors:** Azadeh Eskandari, Thean Chor Leow, Mohd Basyaruddin Abdul Rahman, Siti Nurbaya Oslan

**Affiliations:** 1Enzyme and Microbial Technology Research Centre, Universiti Putra Malaysia, UPM, Serdang 43400, Selangor, Malaysia; gs52501@student.upm.edu.my (A.E.); adamleow@upm.edu.my (T.C.L.); 2Department of Biochemistry, Faculty of Biotechnology and Biomolecular Sciences, Universiti Putra Malaysia, UPM, Serdang 43400, Selangor, Malaysia; 3Department of Cell and Molecular Biology, Faculty of Biotechnology and Biomolecular Sciences, Universiti Putra Malaysia, UPM, Serdang 43400, Selangor, Malaysia; 4Enzyme Technology Laboratory, Institute of Bioscience, Universiti Putra Malaysia, UPM, Serdang 43400, Selangor, Malaysia; 5Department of Chemistry, Faculty of Science, Universiti Putra Malaysia, UPM, Serdang 43400, Selangor, Malaysia; basya@upm.edu.my

**Keywords:** antifreeze protein, thermal hysteresis, ice recrystallization inhibition, antifreeze glycopeptide, psychrophiles, application

## Abstract

Antifreeze proteins (AFPs) are specific proteins, glycopeptides, and peptides made by different organisms to allow cells to survive in sub-zero conditions. AFPs function by reducing the water’s freezing point and avoiding ice crystals’ growth in the frozen stage. Their capability in modifying ice growth leads to the stabilization of ice crystals within a given temperature range and the inhibition of ice recrystallization that decreases the drip loss during thawing. This review presents the potential applications of AFPs from different sources and types. AFPs can be found in diverse sources such as fish, yeast, plants, bacteria, and insects. Various sources reveal different α-helices and β-sheets structures. Recently, analysis of AFPs has been conducted through bioinformatics tools to analyze their functions within proper time. AFPs can be used widely in various aspects of application and have significant industrial functions, encompassing the enhancement of foods’ freezing and liquefying properties, protection of frost plants, enhancement of ice cream’s texture, cryosurgery, and cryopreservation of cells and tissues. In conclusion, these applications and physical properties of AFPs can be further explored to meet other industrial players. Designing the peptide-based AFP can also be done to subsequently improve its function.

## 1. Introduction

Numerous organisms advance exclusive adaptive mechanisms for surviving in exceptionally cold environments. Approximately two-thirds of the earth’s surface contains water, where the average surface temperature of seas and oceans varies within −2 to 30 °C, depending on the latitude [1,2]. In Antarctic and Arctic areas, the temperature of the seawater is constantly sub-zero due to the existence of dissolved salts. These subzero temperatures are very damaging, if not fatal, to polar organisms. As a cold-variation mechanism, antifreeze glycopeptides (AFGPs) or antifreeze proteins/peptides (AFPs) are produced by various organisms for protecting them from freezing injury [3,4].

These peptides and proteins contain ice-binding affinity and are able to reduce the freezing temperature of a solution non-colligatively with a negative influence on its melting temperature, a phenomenon known as the thermal hysteresis (TH). Furthermore, AFGPs can prevent recrystallization of ice where large ice crystals are developed at the expenditure of the smaller ones, and thus avoid the ice regrowth-caused mechanical cell damage. Presently, it is commonly agreed that the AFGPs function through adsorbing their flat ice-binding surfaces over the specific ice crystals’ planes, hence inhibiting or preventing the further growth of ice [5].

Due to special features and functions of these proteins, their capabilities have been utilized in cell biology, food industry, and biotechnology. In this review, we describe the types, features, properties, and characterizations of antifreeze proteins. Furthermore, it also covers the biotechnological study of AFPs via different tools and web servers to easily analyze the features and properties of AFPs. Finally, we try to elevate the role of these proteins/peptides in diverse aspects of industrial, agricultural, and medical proposes.

## 2. Discussion

### 2.1. Properties and Characterizations of AFPs

In general, AFPs can be considered in terms of two features: TH and ice recrystallization inhibition (IRI) features. They belong to a subgroup of ice-binding proteins (IBPs) depressing the hysteresis freezing point known as the ‘thermal hysteresis gap’. Two hysteresis intervals are involved in this TH gap, including a melting hysteresis (MH) interval within melting point (MP) and the equilibrium melting/freezing point, and a freezing hysteresis interval [6,7]. In the solution containing AFP, the temperature gap can be made through irretrievable binding of ice crystals to AFPs and consequent prevention of their growth until decreasing the temperature to the non-equilibrium freezing point [8,9].

Under this condition, the ice crystals burst. Figure 1 shows the characterization of antifreeze proteins from different organisms in nature [10]. IRI is the second function of AFPs [11]. It explains a thermodynamically encouraging procedure where the larger ice grains are created by the smaller ones possessing high internal energy (Figure 2) [12].

AFPs can also bind to the particular ice crystals’ planes within the TH gap, resulting in exclusive ice morphology [13]. For instance, winter flounder and sculpin Type I AFPs bind to the ice pyramidal and secondary prism plane, respectively (Figure 3). Hyperactive AFP from Antarctic bacterium *Flavobacterium frigoris* PS1 (FfIBP) binds to the basal plane and a lemon shape is created, while hyperactive AFPs can bind to the ice crystals’ basal plane [14].

Recently, most research concerning AFPs has focused on exploring, recognizing, purifying, evaluating, and detecting methods, analyzing their mechanisms, and characterization as well as bioinformatics studies to support the potential utilizations of AFPs in different aspects. For instance, ice affinity purification is a classic method for AFPs purification and is used to extract AFPs from the solution [15]. Similarly, AFPs can be purified from cold-acclimated malting barley via ammonium precipitation and chromatography techniques [16]. *Gadus macrocephalus* Type IV AFP (GmAFP IV) and AFP derived from berry were purified within affinity chromatography [17,18].

Experimental methods had been used to assess the mechanism of antifreeze proteins containing site-directed mutation as well as ice-etching [19,20]. As previously mentioned, different types of AFPs could bind specifically to ice planes, as was demonstrated by Knight et al. [21]. Researchers developed a method for ice-etching to identify the ice-affinity plane. Subsequently, fluorescence-based ice plane affinity (FIPA), which fused fluorescent tag or covalent stain with AFPs, could determine the interaction of ice with AFPs plane crystals [20].

High-resolution microscopy as well as immuno-fluorescence could be utilized to reveal the variations in distribution of AFPs [22]. Orbital computational analysis was used to survey the interaction of AFP II with ice lattice that could prevent ice recrystallization [23,24]. Some useful methods could also be applied to analyze the AFPs’ functions. TH is the most popular test for measuring AFP activity. One of the standard methods for measuring the TH of AFP is nanoliter osmometry (Cryoscopy) [25]. Moreover, Knight et al. [25] developed the splat-cooling method for the IRI assay that was later replaced by the sucrose sandwich assay [26]. Then, a new technique to preserve samples was invented through testing under supercooling methods to detect the IRI activity [27,28,29].

Recently, screening and bioinformatics tools were effective and accurate to analyze the diverse types of AFPs. For instance, the hydration structure of AFPs could be analyzed by random tool modeling [30,31], meanwhile the AFPs salvation layer could be analyzed by molecular dynamic (MD) simulations [32]. Additionally, there are some user-friendly website servers that could analyze the secondary structures and easily predict their structural models [33]. Target Freeze and iAFP-Ense software (An Ensemble classifier for identifying of AFP) could be used to predict the presence of pseudo amino acids in AFPs [34,35], while homology modeling could be performed by using other servers [36,37]. Therefore, approaching various aims, experimental studies as well as bioinformatics tools could provide stronger support and further justification of the results besides being time- and cost-saving [38,39].

### 2.2. Origin, Classifications, and Diverse Structures of Antifreeze Proteins (AFPs)

AFPs are one of the main proteins that remain functional in cold temperatures naturally encompass bony fish, insects, plantae, fungi, Nematoda, amphibia, bacteria, and diatoms [11] (Figure 1).

AFGPs are produced by various organisms, categorized into five different classes (AFGP and types I–IV) depending on their sources and structural features [40]. Arctic and Antarctic organisms have been identified to have or express these AFPs in different types and structures (Figure 4) [41]. Besides possessing various structures, they could be categorized based on their capability in binding to ice and preventing or slowing down the ice growth and/or ice recrystallization [42].

The Type I AFP which has a straight α-helix chain with molecular mass of 3.3–4.5 kDa and length of almost 58 Å is abundantly derived from diverse fish species as well as yeasts. It comprises 37 amino acids, where most of the residues are alanine. Furthermore, it involves three repeats of Thr-Ala-Ala-X-Ala-X-X-Ala-Ala-X-X [43].

Besides, the Type II AFP includes the greatest number of non-glycoproteins and thermal hysteresis proteins THPs, and exists in various species, including Atlantic herring, sea raven, and smelt. The structures of the protein are stabilized through disulfide bonds within cysteines, and the general tertiary structures are categorized as two α-helices and two β-sheets. The molecular mass is in the range of 11–24 kDa [44,45].

The Type III AFP exists in eelpout and both Northern and Antarctic zoarcid fish. The molecular mass is in the range of 6.5–14 kDa. These AFPs are classified into two groups in terms of their affinity towards quaternary (QAEs) and sulfopropyl (SP) amines. Type III AFP contains no cysteine residues and no amino acid bias. The two anti-parallel triple-stranded β-sheets make their secondary structures more compact. The three-dimensional (3D) structure of this type of AFPs forms the shape of a spherical or rather a triangular prism with two moderately flat faces. The prism’s side has a key role in ice-binding interaction [14,46,47].

Lately, Type IV AFP has been revealed in *Longhorn sculpin* fish and is wholly different from the other types of AFPs. It is a protein enriched in glutamine and glutamate amino acids. It has the molecular weight of nearly 12 kDa and consists of 108 amino acids, which is much greater than other identified sculpin AFPs. Interestingly, its structure is completely made up of helices with a considerable number of α-helical folds, where the hydrophobic sides are in front of the interior and four amphipathic helices are organized anti-parallel, with the polar sides fronting the solvent [3].

The AFGP is available in the blood sera of Antarctic notothenioid and indicated that this type of AFP contains some repeating residues of (Ala-Ala-Thr)_n_, where the disaccharide β-D-galactosyl-(1,3)--N-acetyl-α-D-galactosamine is linked as a glycoside to the hydroxyl oxygen of the Thr residues. The residues are repeated in the structure of protein for 4–30 amino acids, and proline is rarely involved in the isoforms. The molecular weight is within 2.7–32 kDa. Overall, AFGPs having molecular weights between 20 and 33 kDa are referred to as AFGP 1–4, and that less than 20 kDa are denoted as AFGP 5–8 [48,49].

Even though AFPs are produced by a large group of insects, only a few of the structures are recognized, including the beetles (*Coleoptera*) *Tenebrio molitor*, *Microdera punctipennis, Dicentra canadensis, Rhagium inquisitor*, and *Anatolica polita* [50]. Most terrestrial arthropods (insects, mites, and spiders) [51] express AFPs in the winter. Cysteine is the only amino acid that presents abundantly in some insect AFPs, while other AFPs are rich in serine, threonine, and β-helix structure [51]. The insects AFPs’ unique characteristics are 10 to 100 times more active than AFPs from fish, allowing them to survive in much colder situations [52].

Initially, researchers demonstrated the presence of AFP in winter rye leaves (*Secale cereale*). Later, scientists detected antifreeze protein from wheat (*Triticum aestivum*). Various investigations were conducted on plants’ antifreeze proteins in carrots (*Daucus carota*), ryegrass (*Lolium perenne*), *Solanum tuberosum*, *Solanum dulcamara*, *Forsythia suspense*, *Picea abis*, and *Picea pungens*, among many others [53,54]. Ice-forming inhibition activity exists in overwintering plants merely following their exposure to low temperatures. Ice structuring proteins (ISPs) are involved in numerous types of perennial and overwintering plants. The plant AFPs have less TH but display a robust IRI activity [45].

### 2.3. Applications of Antifreeze Proteins

In recent years, the potential applications of AFPs have become more widespread based on gradual information obtained from identification and characterization of their properties. Mostly, the growth of ice crystals creates major challenges in different fields. AFPs and their dependent analogues are highly utilized to inhibit growth of ice crystals in freezing conditions due to the key biological role of AFPs in low temperatures. Therefore, the current potential applications of AFPs are reviewed.

#### 2.3.1. Industrial Applications of AFPs

##### Frozen Food

Numerous biotechnological and industrial procedures have utilized cold-active biomolecules from organisms modified to create a cold environment. Ice-binding proteins (IBPs), including AFPs and their recombinants, are principally employed in various areas such as medicine and food industries. Fish recombinant AFPs are applied to improve food conservation over freezing [50]. Ice nucleation proteins (INPs) with high activity at the warmest temperature are predominantly isolated from bacteria and have been used in different areas.

Encouraging findings were reported in various food technology investigations on the use of bacterial INPs to increase the ice nucleation temperature, with subsequent decrease in freezing times and the size of ice crystals. Thus, these further enhance the quality of frozen solid food. As a result of creating ice at higher temperatures, a lower energy cost is needed for food freezing. It was also shown that freezing food with less cooling was possible when utilizing biological ice nuclei created by bacteria [50,51]. In this issue, the usage of AFPs can open new opportunities for food and fruit storage due to their special properties. Nonetheless, the in vitro usage of antifreeze proteins could only maintain the tissues superficially. Therefore, the mining of AFPs’ coding genes in plant organisms is crucial to generate fruits which are resistant to the freezing–thawing process. Furthermore, altering the plant genetics can develop food and fruits with higher quality [55].

##### Ice Cream Products

Ice recrystallization is among the difficulties in the ice cream industry. It is initiated by temperature fluctuation over storage, where small-size ice crystals are changed into the larger ice crystals through double freezing. The organoleptic and texture features of ice cream are destroyed by the large size of ice crystals. Hence, AFPs are recommended as an inhibitor for the natural growth of ice in freezing dairy yields. Ice cream products with optimal ice crystal structure play a crucial role in maintaining a creamy and smooth texture. AFPs offer continuity of the ice cream’s smooth structure via inhibition of the crystallization [54,55,56]. Ice cream products including AFPs are now commercially available [57].

##### Frozen, Chilled Fish and Meat

AFPs’ ice recrystallization prevention feature is used in frozen fish and meat. An initial study had used some pieces of bovine meat saturated in a concentrated drain of AFP solution to make it dry [54]. They found that AFP kept the ice crystal size in the frozen meat. However, there was a problem regarding the long saturating time of the meat in the AFP solution (two weeks), which caused deterioration. Thus, AFP solution could be injected to the lambs prior to slaughter to prevent this phenomenon [50].

In other investigations, the impacts of pre-slaughtering AFGPs injection to lambs were evaluated in terms of the quality of lamb meat after thawing [51,57]. The lambs were injected intravenously with the AFGPs from Antarctic fish before slaughtering and after thawing. The impacts of AFGPs on thawing of freezing specimens and meat quality were evaluated. The specimens maintained frozen for 2–16 weeks at −20 °C and were vacuum-packed. The drip loss intensity and ice crystal size were decreased by injecting AFGP at either 1 or 24 h prior to slaughtering. In the lambs receiving 0.01 pg/kg concentration of AFGP, the ice crystals were in the smallest size, particularly when injecting AFGP 24 h prior to slaughtering [57].

Based on these findings, by developing economical and consumer-acceptable techniques, frozen storage damages can be reduced by adding AFGP into the meat before freezing. To extend the products’ shelf lives, frozen storage is needed. Nevertheless, another problem would be initiated regarding freeze-induced denaturizing of the muscle proteins. Using cryoprotectants somewhat prevents this denaturization, where sucrose and sorbitol (4% *w*/*v* each) are commercially utilized as the cryoprotectants. Though, these sugars lead to sweetness, which is non-desirable for some consumers and for certain food products [56,57].

Presently, additives without this drawback are not commercially accessible. AFPs may respond to this problem. Initially, the studies demonstrated the potential effectiveness of Type III AFP in preserving the gel-creating features of fish under chilled and frozen circumstances. Consequently, even the following frozen temperature is misused, AFP (with a comparatively lower AFP concentration) still offers better protection compared to the conservational cryoprotectants. It results in no sweetness, which is a commercial advantage. Through discovering this innovative AFP capability, an exploration into other food uses is encouraged [57,58].

##### Frozen Dough

Since 1960, the frozen dough method has attracted a huge deal of interest to cope with the problems of the short shelf-life of conventional doughs. However, this method can result in the reduced retention capacity and the weakened dough structure. Although utilizing freeze-tolerant yeasts or strong wheat flour could prevent such drawbacks, the prolonged fermentation time has led to the deterioration in the bread texture [59,60].

To overcome this issue, Zhang et al. [60] studied the impacts of adding the protein supplementation of carrot concentrated with 18.3% Carrot AFP (CaAFP), soy protein-isolated (SPI), bovine serum albumin (BSA) on the frozen dough’s fermentation capacity. With the addition of the concentrated protein to the frozen dough, the carbon dioxide (CO_2_) retention capacity was increased, the yeast mortality was decreased, and the delayed crystal creation resulted in a TH phenomenon over the freezing and thawing cycle. The stronger fermentation capacity, and greater CO_2_ retention capacity, were found via the CaAFP-added group compared to SPI- and BSA-added groups [60,61].

Nevertheless, there is no proper yeast candidate for the industrial bakery with optimum gassing capacity in frozen dough. Moreover, it is unlikely that considerable enhancements of this attribute can be provided by classical breeding programs. A study was conducted to express a recombinant gene of antifreeze peptide GS-5 from the polar fish grubby sculpin (*Myoxocephalus aenaeus*), in vitro, and industrial strain of baker’s yeast (*Saccharomyces cerevisiae)* [62]. In both strains tested by producing the recombinant protein, the freezing tolerance was incrementally increased [63].

Recent studies showed that antifreeze proteins utilized in frozen dough could enhance and improve the total gas and gassing rate. Moreover, using high concentrations of AFPs from carrot can increase the quality and texture of frozen dough through keeping the volume of loaf within freezing process [64]. Effects of CaAFP extraction and CaAFPs were investigated on texture properties, thermophysical features, cooking features, and microstructure of frozen white salted noodles. The findings indicated that the texture properties and cooking features of frozen noodles were enhanced with CaAFPs supplementation, even though by extended storage, the improvement was declined [59].

##### Creating Freeze-Tolerant Gels and High-Porosity Ceramics

A gel structure is included in numerous water-containing substances, in which water molecules are held by a polymer network to make an inter-tangled texture possessing good viscosity. Jelly, noodles, boiled eggs, dough, tofu, and cakes are all damaged by the freezing–thawing procedure through forming ice crystals which alter the internal texture [65].

The accumulation of AFP on embryonic ice crystals inhibits the ice blocks’ creation, a preservative for the polymer network and its water-holding feature over the freeze–thaw procedure. Only 0.05–0.1 mg/mL of AFP is needed for the protection of 0.5% (*w*/*v*) agarose gel. AFPs for maintaining the gel structures can be used for different purposes. The “gelatin freezing technique” is expanded to create porous materials, where AFPs made a considerable contribution. Creating such an ice-motivated structure within a gel or other water-containing substance is one of the encouraging applications of AFPs [66].

##### Creating Anti-Ice Surfaces via AFPs on Aluminum

AFPs usage is an environmentally friendly method for anti-frosting or anti-icing, where AFP is used to coat the metals, which was stimulated by trehalose. Frost or ice development results in considerable safety and economic issues in different applications [67]. These issues include the burst power lines, reduced wings’ lifetimes in aircrafts, and also the inhibition of air circulating in freezers and refrigerators [68]. Ice difficulties are worked out through common coating of hydrophilic polymer combined with ZrO_2_ or BaO_2_, or via other cryoprotectants such as antifreeze proteins, polyols, and sugars [69].

Many industrial usages of AFPs are made possible by developing the fusion proteins comprising AFPs and other metal-binding peptides. Proteins can be simply denatured by temperature. Definitely, six days after coating, the *Chaetoceros neogracile* AFP Cn-AFPs immobilized on aluminum were denatured to some extent. In industrial applications, AFPs’ denaturation must be prohibited while maintaining their stability. Trehalose is a famous sugar used for preventing the denaturation of proteins as well as stabilizing their structures under freezing circumstances by forming the various hydrogen bonds between polar amino acids and the hydroxyl groups of trehalose. Super cooling points of AFPs are measured as the TH activity, that is related to the main function of AFPs [67,70].

Recombinant AFP significantly reduced the super cooling point of the solution compared to water. Further trehalose coating dramatically increased the immobilized protein’s stability [67]. The super cooling point of the mutant Cn-AFP binding peptide (Cn-AFP (BP-Cn-AFPG124Y-Al)) was not significantly influenced by trehalose coating. Therefore, protein durability is obviously improved on metal surfaces by trehalose coating. Although large quantities of ice collected on the surface of a traditional hydrophilic-coated aluminum, ice and frost were not created over the Tre-ABP-Cn-AFPG124Y-Al surface.

The advantages of this method for industrial utilizations are: Firstly, recombinant ABP-Cn-AFP proteins are feasible for production on an industrial scale. It is an environmentally friendly structure since the recombinant proteins’ production is bio-based; secondly, preparation of aluminum (Al) surface coating with antifreeze proteins was completed quickly through a one step-dipping technique lacking complex surface alteration. ABP-fused AFPs can maintain the proper orientation of the AFP on a surface, allowing the use of their full anti-icing features. Most prominently, AFPs impede ice creation on the surface of Al in comparison with the bare Al and the usual hydrophilic Al coatings. This impact is caused by the TH activity of Cn-AFP and capability in reducing the freezing point [67,71,72,73].

#### 2.3.2. Medical Applications of AFPs

##### Anti-Infective (Anti-Virulence) Properties of AFP

Bacterial biofilms are a major concern in clinical medicine, with more than 400,000 estimated annual catheter-related bloodstream infections in the U.S. alone. Current treatments are unable to clear most of these catheter-related infections. Thus, treatment is restricted to device removal and replacement, which further increase the risk of re-infection and treatment costs. Described herein is a new anti-infective class that is effective against infectious agents such as bacteria, viruses, fungi, and protozoa [74].

The anti-infective agents include antifreeze proteins and peptides from a variety of organisms in addition to the synthetic proteins and peptides whose amino acid compositions are derived from antifreeze proteins, such as an arthropod antifreeze protein. Nevertheless, antifreeze glycoprotein and glycopeptides from *Ixodes scapularis* tick can also contribute to the anti-infective activity [75]. They are useful against bacteria, viruses, fungi, and protozoa, in the body and on the skin, by reducing infection, by preventing its occurrence, reducing the extent to which it occurs, or treating an existing infection. The *Ixodes scapularis* tick antifreeze glycoprotein (IAFGP) binds to bacteria while altering their microbial biofilm creation in vitro (Figure 5). IAFGP can bind with *Serratia marcescens* (SM), *Escherichia coli* (EC), *Pseudomonas entomophila* (PE), *Staphylococcus aureus* (SA), and *Listeria monocytogenes* (LM) [76,77].

##### Cryomedicine (Cryopresevation/Cryosurgery)

Mainly, cryomedicine is related to cryosurgery and various cryopreservations. Cryopreservation is a necessary stage in different biomedicine regions such as gene therapy, tissue engineering, and tissue preservation screening of drugs. To achieve these aspects, new platforms and reagents with antifreeze as well as compatible osmoprotective properties have been expanded recently for cryopreservation [11].

##### Cell and Tissue Cryopreservation

Generally, destructive ice creation and recrystallization take place both outside and inside of cells over freezing procedures. A two-step freezing technique can solve these problems, where the cells are initially chilled at 0 °C in a preservation fluid. During the first phase, only the material surrounding the cell is frozen, where dehydration is performed. At the second stage, the cells are frozen completely with liquid nitrogen (−196 °C). Previous dehydration reduces the formation of ice crystals between the cells in this phase [78].

In general, a programmable freezer is used in this technique, where the whole process will take about 2–3 h. Vitrification is another procedure, where the cells are frozen quickly to reach −196 °C with greatly concentrated (1–3 M) cryoprotectants (e.g., dimethyl sulfoxide (DMSO), acetamide, propylene glycol) which penetrate the cells [79]. This technique can inhibit ice crystal formation both outside and inside these cells. These cryoprotectants are fully toxic in a liquid phase. Therefore, the cells need to be frozen rapidly to cease cell death prior to freezing after mixing the detergents/solvents. Moreover, the cells are damaged over the thawing procedure by both the detergent toxicity and ice recrystallization factors.

The detergents applied in the two-step freezing technique are also toxic, however, they are much more diluted compared to those utilized in vitrification. Within cold-tolerant organisms, the function of AFPs combined with different organic and inorganic materials, including glycerol, glucose, ions, lipids, mineral salts, peptides, and carbohydrates. Therefore, utilizing optimal combinations of these substances can result in the maximized AFP performance differing between the usages [80,81].

AFPs are particularly effective in frozen solutions for inhibiting ice recrystallization. Initially, in 1969, a study mentioned and highlighted the crucial functions of the proteins in freeze-tolerant organisms. In vitro examination of red blood cells (RBCs) had illustrated the effects of AFP on the recovery of cryopreserved cells [82], which could regularly endure cool conditions and consequently be injured by ice crystal growth over warming. Survival of RBCs that were cryopreserved in hydroxyethyl starch solution could be enhanced by moderately low concentrations of AFPs from winter flounder (*Pseudopleuronectes americanus*). Comparatively high concentrations of these proteins were furthermore operative in preventing the extracellular ice recrystallization for RBCs [83]. Previously, one controlled test showed damaging conditions of tumor cells regarding to their exposure to AFP [84].

Studies showed that the injury gained at various phases over the cryopreservation process may result in the follicular depletion. Therefore, preservation within sub-zero conditions would help keep ovarian function. The results indicated the cryoprotective impacts of antifreeze proteins, particularly a high concentration of LeIBP (AFP), on ovarian tissue over the warming/vitrification procedure. The results offered a basis for further investigation on the mechanisms and impacts of AFPs on human ovarian tissues [78,85].

Moreover, other studies showed that AFGP8 also had a positive effect on bovine oocytes against cold condition injuries during the vitrification process [86]. Type III AFP had useful effects for oocytes and ovarian transplantations [87,88,89], however it was less effective in preventing bovine oocytes cryoinjuries [90].

In the early twentieth century, marine AFPs were initially utilized to protect the cell membranes of oolemma at hypothermic temperatures via AFGP from Antarctic and Arctic fish. Although most cryopreservation trials using AFPs had confirmed that adding AFPs could enhance the freezing and thawing sperm viability, numerous reports indicated no beneficial effect despite the freezing technique, storage temperature, and biological specimen [91]. AFPs have been utilized in the cell lines’ cryopreservation, as additives to conventional freezing medium to decrease the high quantity of cytotoxic cryoprotectants (CPAs) and decrease the freezing loss. Sperm cell was one type of the cells which used this technique and the AFPs were examined for their cryopreservation [92].

Fish AFPs were investigated previously, for their effectiveness when supplemented in the sperm preservation media for various organisms at freezing and chilling steps. In ram, bovine, chimpanzee, buffalo, and fish, sperm quality was enhanced using cryopreservation and chilled storing through AFPs [91]. Another study also reported on the availability of the Type I AFP-exhibited positive effect on high fertility of bovine sperm within the freeze–thaw process. In two other investigations, motility and integrity of plasma membrane in semen of (Nili-Ravi) buffalos were improved when exposed to Type III AFP and AFGP during the freezing–thawing process [93,94]. Type III AFP might also have beneficial effects on chimpanzee sperm. As compared to Type I and III AFPs, Type III had better function during the cryopreservation of carp *Cyprinus carpio* sperm [95].

It has been demonstrated that Type I and III AFPs had cryoprotectant effects on the embryonic tissues in zebrafish [96] as well as the liver and heart tissue in rats [97,98]. Amir et al. [97] conducted cryopreservation of heart tissues in rat abdomen successfully in 24 h [99]. In addition, Type I AFPs could be used as an adjuvant for cryosurgery of rat prostate and liver tumors, with important effects on those tissues [100].

##### Organ Transplantation

Cryoprotective agents (CPAs) are chemical substances which prevent ice crystal formation in zero to sub-zero conditions in organ transplantation. Current CPAs consist of small molecules (frequently known as penetrating CPAs), AFPs, and synthetic polymers [76]. Presently, organ transplantation is a substantial remedy for giving high-level organ failure, which is based totally on the life situation, value-effectiveness and survival. Tissue loss in organs resulted from the inadequacy of proper maintenance materials [65].

In organ maintenance, proper solutions and materials are required for organ stabilization and blood elimination. Currently, various components are used as CPAs, encompassing ethylene glycol, 1,2-propanediol, DMSO, formamide, glycerol, lactose, sucrose, and D-mannitol. These components cause intense cell death due to their high toxicities to organs. Therefore, the massive removal of CPAs upon warming is needed before the transplantation process [76].

AFPs and some synthetic polymers are special CPAs that prevent ice crystal formation at low temperatures. Despite the fact that AFPs that are derived naturally from fish, insects, and plants play an important role in prevention of ice crystal formation, the low purity and high cost of these proteins are the major drawbacks. Moreover, AFPs usage in organ tissues’ preservation may possess immunogenicity to induce immune responses [101].

#### 2.3.3. Agricultural Applications

##### Effects of AFPs in Frost Protecting Crop Plants

The AFPs’ capability in thermal hysteresis prevents the damage caused by common minor frost measures in early autumn and late spring in the frost-sensitive crop plants. Therefore, the expression of the high-activity AFPs permits them to continue the freezing at nearly −5 °C. Over two decades, the effectiveness of this concept was examined in various investigations producing transgenic plants (*Arabidopsis thaliana* and several crop plants) which expressed different AFPs [102].

Originally, AFPs from fish were used in these assessments but the insect AFPs were found to exhibit higher activity. Then, they were used for plant transformation investigations. Transgenic plants produced in some studies showed the enhanced cold-tolerance of 1–3 °C in comparison with their wild-types. However, no studies with transgenic plants have found an adequate level of protection as yet. Progress up to this point shows the possibility of obtaining more significant results [103,104].

Research shows that AFPs can be applied for agriculture and aquaculture products; however, those methods have not been used at present. Previously, the Type I AFP expressed in the winter flounder was evident to prevent the growth of ice lattices in cold conditions, where such observation had led to the improved economic situation of dead fish [105]. Similarly, some AFPs from transgenic plants have potential to increase the growing geographical areas by expanding the seasons of crop growing, such as potato, canola leaves, and wheat [105]. However, various studies have reported on the limited effect of AFPs on plants crops, therefore, further studies are much needed. Application of various AFGPs derived from different types has been shown in Table 1.

## 3. Conclusions and Future Perspectives

The ability of organisms to tolerate cold conditions is a natural phenomenon in nature. The development of AFPs reveals a quintessence to comprehend this self-protection circumstance. Over the last decade, our understanding on the utilization of AFP have confirmed the promising usages of this protein with its specific function in different medical and industrial fields. Developments in both structural and physicochemical biological and genetic approaches significantly contribute and allow us to obtain an unprecedented perception towards the mechanisms of membrane protection and ice-binding of AFP. The useful applications of AFPs have been proven in numerous investigations. However, the time-consuming purification process of AFPs has limited the huge usage and functionality of these beneficial biomolecules in various applications.

This review focused on potential properties of AFPs with different sources, structures, and characteristics. Further amounts of AFP obtained via large-scale methods will permit its functional developments in medical and industrial fields. Support from government agencies is needed for the medical application of AFPs regarding its mutagenicity, carcinogenicity, and toxicity. Hence, the specimen should be made in a facility satisfying the regulation of good fabricating practice. The necessary data will be provided by more assessments on abundant aspects of AFP to solve these problems.

## Figures and Tables

**Figure 1 biomolecules-10-01649-f001:**
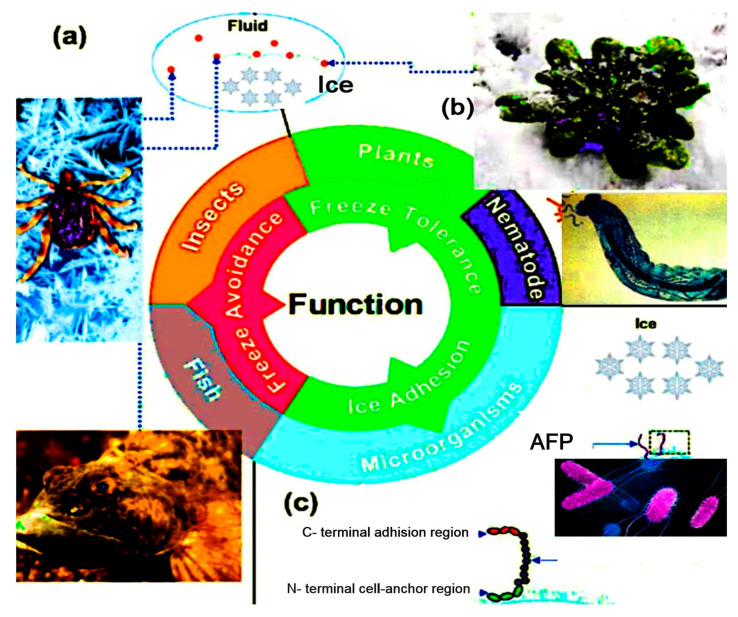
The natural properties of antifreeze proteins (AFPs) (thermal hysteresis and ice recrystallization inhibition) in various organisms in nature. (**a**) For freezing inhibition, AFPs can block forming of ice crystals in fish and insects by lowering down the freezing points in body fluids. (**b**) In plants and nematode, freeze-tolerating is carried out by binding the AFPs to the surface of ice and prevent it from becoming larger ice crystals. (**c**) AFPs of different microorganisms can adhere to the ice surface, like *Marinomonas primoryensis*, and inhibit the formation of ice crystals. Adhesion of AFPs to ice can be done from three regions: C terminal, N terminal, and intermediate repeat [10].

**Figure 2 biomolecules-10-01649-f002:**
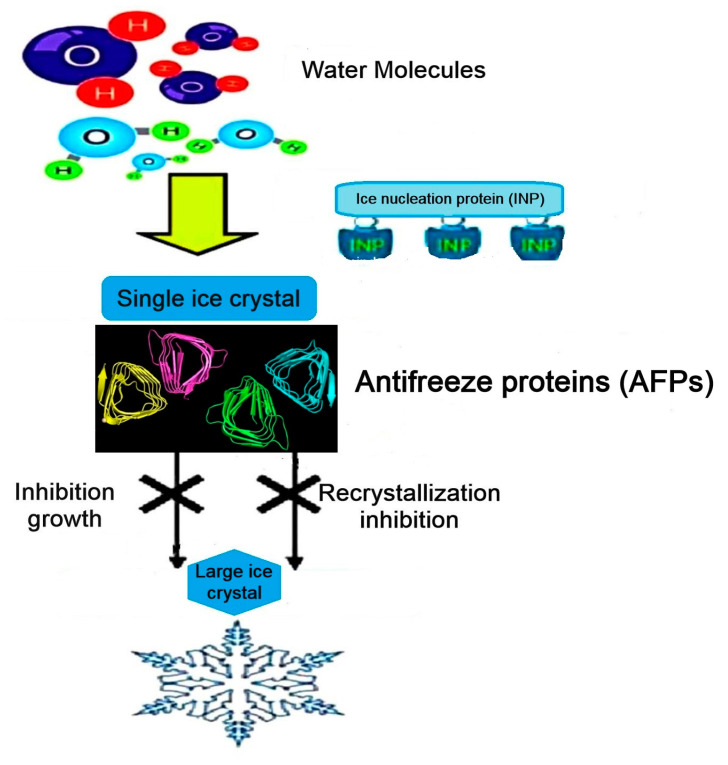
Ice recrystallization inhibition function of AFPs [12].

**Figure 3 biomolecules-10-01649-f003:**
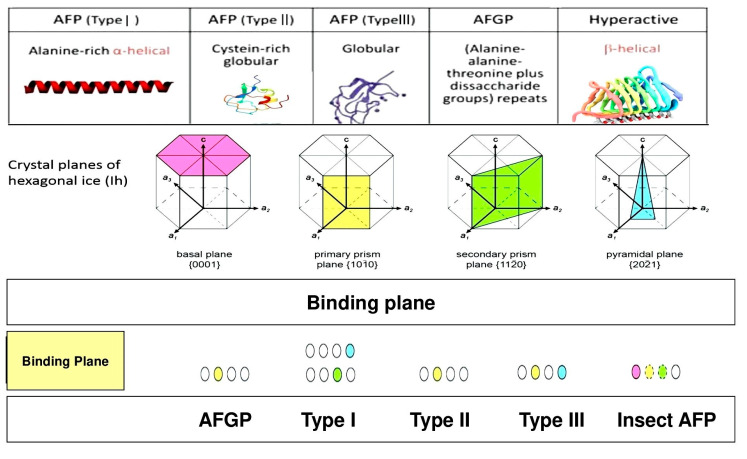
Structural properties of AFPs based on binding to ice crystal planes, which are widely different due to the types and structure of AFPs. Type I AFP has a primary prism shape of the ice plane. Other types can form basal plane, secondary prism, and pyramidal shapes once they interact with ice/water [14].

**Figure 4 biomolecules-10-01649-f004:**
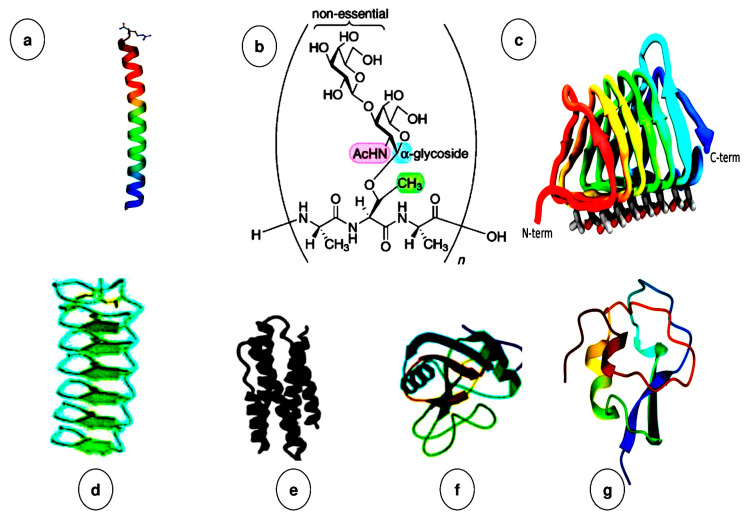
Antifreeze protein structures. Fish AFPs (**a**,**b**,**e**–**g**) and two AFPs from insects (**c**,**d**) with α-helices, β-strands and coil. (**a**) Type I AFP from winter flounder, (**b**) antifreeze glycoprotein (AFGP) with left-handed helix, (**c**) insect AFP from *Spruce budworm*, (**d**) AFP from *Tenebrio molitor*, (**e**) Type IV AFP with 4 bundle helices with unknown characteristics, (**f**,**g**) non-repetitive AFPs: (**f**) Type II AFP from sea raven (The protein data bank (PDB) ID: 2AFP) and (**g**) Type III AFP from ocean pout [41].

**Figure 5 biomolecules-10-01649-f005:**
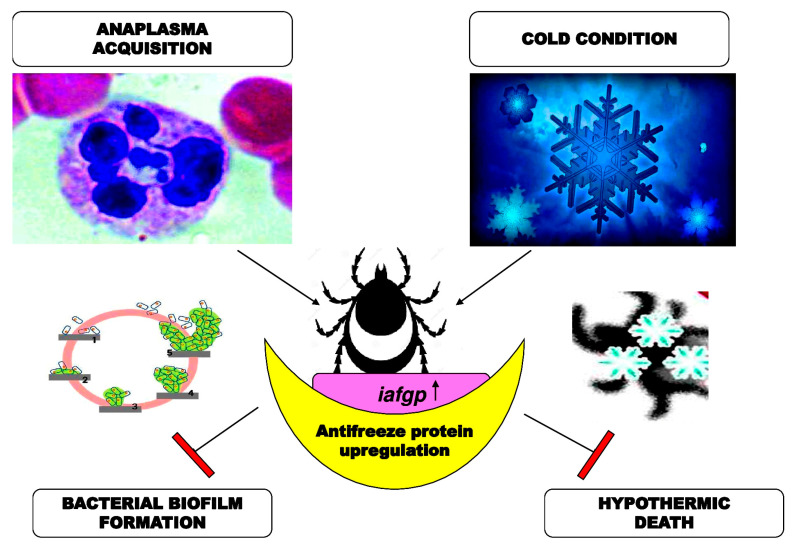
Functions of *Ixodes scapularis* antifreeze glycoprotein (IAFGP) as an anti-virulence element against various microbes, such as the methicillin-resistant *Staphylococcus *aureus (MRSA)** [75].

**Table 1 biomolecules-10-01649-t001:** Applications of diverse antifreeze proteins, peptides/glycoprotein (AFGPs) from different sources.

Industrial Applications	Organism Source (Protein Type)	Main Functions
Food	Fish (type III)	Applied as recombinant protein to improve milk fermentation, especially for the storage of frozen yogurt and ice cream
	Plant (type III)	Screened for proteolytic and biolytic activities to obtain a heat-stable protein
	Fish (type I)	Applied as recombinant protein to use in frozen food such as meat
Medicine	Fish (type I)	Cryopreservation of rat organs
	Fish (type III)	Cryopreservation of mammalian cells
	Fish (type I)	Cryopreservation of red blood cells
	Fish (type I)	Cryopreservation for subcutaneous tumor cells
Agriculture	Bacteria	As biofertilizer to increase plant growth at cold temperature
	AFGP	Inhibition of ice nucleating activity in *Erwinia herbicola* (a bacterium)

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
