# Peer review of "Antifreeze Proteins and Their Practical Utilization in Industry, Medicine, and Agriculture"

_biomolecules, 2020, doi:10.3390/biom10121649_

Round 1

Reviewer 1 Report

The authors describe the practical utilization of antifreeze proteins, in addition to the structural diversity, and functional properties of AFPs from various organisms (fish, yeast, plants, bacteria and insects.), and analytical methods for TH activity, AFP-bound ice plane, ice-recrystallization inhibition, and bioinformatics approaches. The abilities of AFPs have been studied for practical applications, such as maintaining quality of frozen food, low temperature-preservation of cells, grafting cold-tolerant to plant. Therefore, the manuscript significantly advances our understanding of potential of AFPs in the field of industry, medicine and agriculture.

The manuscript is written carefully, but there are some mistakes in selection and/or numbering of references. Additionally, I feel there are some places where input from a native language speaker would improve the grammar.

Page 1, line 28: Is “Antifreeze glycopeptides” “Antifreeze proteins”?

Page 2, line 76: Winter flounder and sculpin type I AFPs bind to the ice pyramidal and secondary prism plane, respectively.

Page 2, line 9: What organism expresses FfIBP? I suggest adding source, for example, “FfIBP (from Flavobacterium frigoris)” or “FfIBP (from Antarctic bacterium)”.

Page 4, line 101: It is not clear what the sentence “Type I of AFP has basal shape of ice plane. Other types of AFPs can inform hexagonal plane or secondary prism place once AFPs interact with ice/water” means.

Page 4, line 103: Ref. 15 should be ref.14.

Page 4, line 108: Ref. 16 should be ref.15.

Page 4, line 111: Ref. 17 should be ref.16.

Page 4, line 112: Ref. 18 should be ref.17.

Page 4, line 114: Ref. 19, 20, and 21 should be ref.18, 19, 20, respectively.

Page 4, line 115: Ref. 25 describes ice recrystallization. Ice-etching method is explained in the article, Knight, C. A.; Cheng, C. C.; and DeVries, A. L.; Adsorption of alpha-helical antifreeze peptides on specific ice crystal surface planes, Biophys J. 1991, 59, 409–418.

Page 4, line 118: Ref. 22 and 23 should be ref. 20 and 47, respectively.

Page 4, line 117: The method combining fluorescent tag or covalent stain with AFPs is called fluorescence-based ice plane affinity (FIPA). See ref. 47.

Page 4, line 121: Ref. 24 should be ref. 23.

Page 4, line 125: I feel ref. 15 is unnecessary.

Page 6, line 176: It is not clear what the sentence “the Ala position is mainly alanine.” means.

Page 6, line 179: Type II AFP has been identified in bacteria? It is recombinant expression?

Page 6, line 187: I feel ref.63 is not appropriate.

Page 6, line 195: the sentence is incomplete.

Page 6, line 196: I feel ref. 47 and 48 are not appropriate.

Page 6, line 203: ref 45 and 49 are appropriate?

Page 6, line 215: What is “ISPs”?

Page 7, line 239: “(type I)” is miswrite?

Page 8, line 303: “the yeast mortality yeast” should be corrected to “the yeast mortality”.

Page 9, line 350: ref. 71 should be substituted with ref. 67.

Page 13, line 497: There are some mistakes in Table 1. “type 1” is “type I”?,  “type 3” is “type III”?, and “Yeast (type 1)” is “Fish (type I)”?

Author Response

                                                                                               27th November 2020

To:

Assistant Editor,

Biomolecules

Dear Mr. Mars Tan

MANUSCRIPT REVISION FOR BIOMOLECULES-976079

Enclosed is the soft copy of a review article entitled “Antifreeze Proteins and Their Practical Utilization in Industry, Medicine and Agriculture by Eskandari et al. as the manuscript revised for Biomolecules-976079. Kindly refer to the APPENDIX for the point-to-point responses in addressing reviewers’ comments.

Thank you for your kind consideration of this manuscript.

Yours sincerely,

Siti Nurbaya Oslan (PhD),

Department of Biochemistry,

Faculty of Biotechnology and Biomolecular Science,

Universiti Putra Malaysia,

43400, Serdang, Malaysia.

Email: snurbayaoslan@upm.edu.my

Tel: +603-9769 6710

APPENDIX

REVIEWER 1

Point 1:

The manuscript is written carefully, but there are some mistakes in selection and/or numbering of references. Additionally, I feel there are some places where input from a native language speaker would improve the grammar.

Response 1:

The reference numbering has been edited accordingly in the text from page 2 onwards. The manuscript has been proofread by 3 people who have written and published numbers of scientific papers. One of them is the Chief Executive Editor of Pertanika Journals.

Point 2 and responses

Page 1

Line 28: Is “Antifreeze glycopeptides” “Antifreeze proteins”? (Response: has been changed to “AFPs”, Line 26)

Page 2

Line 76: Winter flounder and sculpin type I AFPs bind to the ice pyramidal and secondary prism plane, respectively. (Response: Line 82,83)

Line 9: What organism expresses FfIBP? I suggest adding source, for example, “FfIBP (from Flavobacterium frigoris)” or “FfIBP (from Antarctic bacterium)”. (Response: Line 84)

Page 4

Line 101: It is not clear what the sentence “Type I of AFP has basal shape of ice plane. Other types of AFPs can inform hexagonal plane or secondary prism place once AFPs interact with ice/water” means. (Response: “Type I of AFP has primary prism shape of ice plane. Other types can form basal plane, secondary prism and pyramidal shapes once interact with ice/water”, Line 88-90)

Line 103: Ref. 15 should be ref.14. (Response: Ref. 14, Line 90)

Line 108: Ref. 16 should be ref.15. (Response: Ref. 15, Line 95)

Line 111: Ref. 17 should be ref.16. (Response: Ref. 16, Line 96)

Line 112: Ref. 18 should be ref.17. (Response: Ref. 17, Line 97)

Line 114: Ref. 19, 20, and 21 should be ref.18, 19, 20, respectively. (Response: Ref. 19,20, Line 99)

Line 115: Ref. 25 describes ice recrystallization. Ice-etching method is explained in the article, Knight, C. A.; Cheng, C. C.; and DeVries, A. L. Adsorption of alpha-helical antifreeze peptides on specific ice crystal surface planes, Biophys J. 1991, 59, 409–418. (Response: This new reference by Knight et al., in 1991, is substituted with the previous one as reference No.21. Line 100)

Line 118: Ref. 22 and 23 should be ref. 20 and 47, respectively. (Response: Ref. 20, Line 103)

Line 117: The method combining fluorescent tag or covalent stain with AFPs is called fluorescence-based ice plane affinity (FIPA). See ref. 47. (Response: “Subsequently, combining fluorescent tag or covalent stain with AFPs that is called fluorescence-based ice plane affinity (FIPA), can determine interaction of ice with AFPs plane crystals [20]”, Line 101,102,103)

Line 121: Ref. 24 should be ref. 23. (Response: Ref.23,24 Line 106)

Line 125: I feel ref. 15 is unnecessary. (Response: the ref.15 was removed and replaced with ref. 25, Line 108).

Page 6

Line 176: It is not clear what the sentence “the Ala position is mainly alanine.” means. (Response: “It comprises 37 amino acids, which most of the residues are alanine. Furthermore, it involves three repeats of Thr-Ala-Ala-X-Ala-X-X-Ala-Ala-X-X,” Line: 138,139,140).

Line 179: Type II AFP has been identified in bacteria? It is recombinant expression? (Response: Yes, it was referred to the recombinant expression, thus we have decided to remove it from type II,” Line: 142,143).

Line 187: I feel ref.63 is not appropriate. (Response: ref. 63 was removed)

Line 195: the sentence is incomplete. (Response: “Greatly, the structure of a protein is completely helices with a considerable number of α-helical folds which the hydrophobic sides are in front of the interior and four amphipathic helices organized anti-parallel with the polar sides fronting the solvent,” Line: 158).

Line 196: I feel ref. 47 and 48 are not appropriate. (Response: ref. 47 and 48 were removed and replaced with ref.3,’ Line: 158)

Line 203: ref 45 and 49 are appropriate? (Response: ref. 45 was removed and ref. 48 was substituted but 49 was maintained to cite AFGP 5-8 ,” Line: 165)

Line 215: What is “ISPs”? (Response: Ice structuring proteins (ISPs),” Line: 178)

Page 7

Line 239: “(type I)” is miswrite? (Response: “(type I)” was removed,” Line: 195)

Page 8

Line 303: “the yeast mortality yeast” should be corrected to “the yeast mortality”. (Response: was replaced with “the yeast mortality”, Line: 254)

Page 9

Line 350: ref. 71 should be substituted with ref. 67. (Response: ref. 71 was replaced with ref. 67 as suggested by the reviewer,” Line: 303)

Page 13

Line 497: There are some mistakes in Table 1. “type 1” is “type I”?,  “type 3” is “type III”?, and “Yeast (type 1)” is “Fish (type I)”? (Response: the suggested amendments were done in Table 1,”

REVIEWER 2

The English editing due to grammars and sentences were corrected properly.

Response 1:

The manuscript has been proofread by 3 people who have written and published numbers of scientific papers. We hope the revised manuscript possesses better English explanatories which sufficiently allow the readers to comprehend the manuscript in detail.

Reviewer 2 Report

Although there is potential surely for a thorough review of potential industrial and other uses for AFPs, the English is sufficiently poor in this paper that as a review it is inadequate.

Much more care is needed with sentences and meaning, even with concepts as simple as thermal hysteresis and ice binding.

Author Response

(The authors gave the same response as above.)

Round 2

Reviewer 2 Report

This review is well written and timely. Progress has been relatively slow at using AFPs in industry at scale and this review will be a fine starting point for workers in the field to see where we are at.